# Advancements in Remote Alpha Radiation Detection: Alpha-Induced Radio-Luminescence Imaging with Enhanced Ambient Light Suppression

**DOI:** 10.3390/s24123781

**Published:** 2024-06-11

**Authors:** Lingteng Kong, Thomas Bligh Scott, John Charles Clifford Day, David Andrew Megson-Smith

**Affiliations:** HH Wills Physics Laboratory, Interface Analysis Centre, School of Physics, University of Bristol, Tyndall Avenue, Bristol BS8 1TL, UK; t.b.scott@bristol.ac.uk (T.B.S.); john.day@bristol.ac.uk (J.C.C.D.); david.megson-smith@bristol.ac.uk (D.A.M.-S.)

**Keywords:** alpha radiation detection, radio-luminescence imaging, alpha fluorescence, long-distance monitoring

## Abstract

Heavy nuclides like uranium and their decay products are commonly found in nuclear industries and can pose a significant health risk to humans due to their alpha-emitting properties. Traditional alpha detectors require close contact with the contaminated surface, which can be time-consuming, labour-intensive, and put personnel at risk. Remote detection is urgently needed but very challenging. To this end, a candidate detection mechanism is alpha-induced radio-luminescence. This approach uses the emission of photons from radio-ionised excited nitrogen molecules to imply the presence of alpha emitters from a distance. Herein, the use of this phenomenon to remotely image various alpha emitters with unparalleled levels of sensitivity and spatial accuracy is demonstrated. Notably, the system detected a 29 kBq Am-241 source at a distance of 3 m within 10 min. Furthermore, it demonstrated the capability to discern a 29 kBq source positioned 7 cm away from a 3 MBq source at a 2 m distance. Additionally, a ‘sandwich’ filter structure is described that incorporates an absorptive filter between two interference filters to enhance the ambient light rejection. The testing of the system is described in different lighting environments, including room light and inside a glovebox. This method promises safer and more efficient alpha monitoring, with applications in nuclear forensics, waste management and decommissioning.

## 1. Introduction

### 1.1. Alpha Particles

Alpha particles consist of two protons and two neutrons bound together into a particle identical to a helium-4 nucleus. They are commonly emitted by the decay of heavy nuclei, specifically those with atomic numbers greater than 82 [1].

Alpha decay is a nuclear transformation process in which an unstable atom releases an alpha particle to become a different element. An example is the decay of americium-241 (^241^Am), which transforms into neptunium-237 (^237^Np) as described by the reaction:(1)$${\;}_{95}^{241}\mathrm{Am}\to_{93}^{237}\mathrm{Np}{+}_{2}^{4}\mathrm{He}$$

As alpha particles traverse a medium, they impart energy to the electrons they encounter, which leads to a gradual deceleration of the alpha particles until they stop. The concept of ‘mean range’ is used to describe the average penetration depth of alpha particles in a material, specifically the thickness needed to attenuate the number of alpha particles to half their original quantity. For alpha particles with an energy of 5.5 MeV, such as those emitted by ^241^Am, the mean range in dry air at standard temperature and pressure is around 4 cm [2].

### 1.2. Damage to Humans from Alpha Particles

Radiation damage in biological systems can occur directly or indirectly. Direct effects involve ionisation interactions close to vital biomolecules, whereas indirect effects predominantly result from radiolysis of water, which comprises 70–80% of a cell. The ionisation of water molecules leads to the formation of unstable charged molecules and subsequent dissociation into smaller ions and free radicals. These highly reactive species can further produce secondary reactive chemicals, potentially interacting with and damaging critical organic molecules such as DNA and RNA [3].

Alpha emitters, when ingested, can inflict substantial DNA damage due to the high linear energy transfer (LET) characteristic of alpha particles. This was tragically exemplified in the fatal poisoning of Alexander Litvinenko with polonium-210 [4]. High-LET radiation like alpha particles deposits significant energy over a short track length, leading to dense ionisation along their path and thereby causing pronounced biological damage [5].

Moreover, alpha radiation can induce a broader biological impact than its direct pathway through a phenomenon known as the ‘bystander effect’, where even non-irradiated cells exhibit damage as a response to neighboring irradiated cells [6].

### 1.3. Challenges in Nuclear Industry and Forensics

The nuclear industry routinely contends with radioactive contamination from fuel cycle by-products [7]. Identifying and characterising sources of contamination is a crucial step during the decommissioning of nuclear facilities. Given that many industrial applications involve actinides—predominantly alpha emitters [8]—the detection of alpha particles is vital for [9]:Preventing radiological terrorist attacks.Managing both intentional and accidental radiological contaminations, along with subsequent forensic analysis.Safeguarding public health and the environment from alpha radiation during decommissioning processes.Conducting remote assessments of nuclear waste conditions.

Currently, commercial hand-held alpha radiation monitors like the Radhound with an alpha probe [10] necessitate close-range scanning of potential sources, typically within millimeters. Conventional detection methods not only place personnel in harm’s way, with increased risks of contamination and radiation exposure but are also labour-intensive and may be impractical in hazardous areas due to safety constraints [9].

### 1.4. Alpha-Induced Radio-Luminescence

In this study, we focus on a technique that bypasses direct detection of short-range alpha particles, opting instead for imaging alpha emitters from a distance by capturing alpha-induced radio-luminescence (RL). Alpha particles, with their dual positive charges, are highly efficient at ionising molecules as they pass through air. This ionisation process releases secondary electrons, which in turn, excite the surrounding nitrogen molecules. As these excited nitrogen molecules return to their ground state, they emit photons. These photons are capable of traveling several kilometers [11,12].

Air RL is predominantly a result of the excitation of nitrogen’s second positive (2P) and first negative (1N) systems, specifically transitions from $C^{3}\Pi_{u}\to B^{3}\Pi_{g}$ and $B_{2}\Sigma_{u}^{+}\to X^{2}\Sigma_{g}^{+}$, respectively. The resulting nitrogen RL spectrum is composed of various broad bands, each 2–3 nm wide. These bands stem from the vibrational and rotational motions within the molecular nuclei that alter the electronic energy states [13].

The RL emission mainly falls within the 300–400 nm range. Notably, the primary emission peaks occur at 315.9 nm (accounting for 10.1% of the total emission), 337.1 nm (25.7% of total emission), and 357.7 nm (17.3% of total emission) [14], as depicted in Figure 1.

The discovery of the alpha RL phenomenon dates back to 1903 when W and M Huggins observed faint light emanating from radium bromide, with a spectrum corresponding to nitrogen [15]. In the 21st century, it has emerged as a promising method for the long-range detection of alpha emitters, heralding a new era in radiation monitoring techniques.

However, the intensity of alpha RL is quite low. A single 5 MeV alpha particle typically generates approximately 100 photons in air [12]. This photon yield is dwarfed by the ambient light present even at night, which can be orders of magnitude greater than the light emitted by alpha RL [16]. This disparity poses significant challenges for detector sensitivity and strategies to effectively block ambient light.

### 1.5. Previous Studies of Alpha Detection

Research into long-range alpha detection via alpha-induced radio-luminescence (RL) can be broadly categorised based on the type of detector employed: point detectors, such as photomultiplier tubes (PMTs) [12,17,18,19,20,21,22], and flame detectors [23], versus camera systems equipped with charge-coupled devices (CCDs) [16,18,24,25,26,27,28].

Point detector systems, like PMTs, are highly sensitive to the RL from alpha radiation, even at low activity levels, due to the photon multiplication process inherent to these devices. Selecting the appropriate photocathode material for a PMT can also enhance its UV light sensitivity and decrease its sensitivity in the visible region, simplifying the challenge of ambient light filtration. Nonetheless, to precisely pinpoint alpha sources over extensive areas, PMTs typically require a narrow field-of-view lens system paired with a scanning mechanism [12]. This scanning process can be time-consuming, and the spatial resolution is contingent on the lens system’s field of view, thus complicating the design further.

Conversely, CCD systems excel in resolution. Superimposing the high-resolution alpha RL signal onto a visible light image allows for the accurate determination of the location and spread of alpha contamination. Additionally, CCDs eliminate the need for extensive scanning, reducing operational time and simplifying lens design. However, the broad spectral sensitivity of CCDs (ranging from 200 nm to 1100 nm) makes them susceptible to ambient light interference; therefore, most of the previous experiments using CCD cameras were carried out in a dark environment.

It is important to highlight that advances such as PMT arrays can offer resolutions superior to that of single PMT [18]. Moreover, intensified CCDs (ICCDs) with CsTe photocathodes, which are exclusively sensitive to UV light, can facilitate ambient light exclusion [29], thus broadening the potential for ambient light adaptation in alpha detection systems.

## 2. Material and Methods

### 2.1. Alpha Sources

Two alpha sources are used for these experiments:A 29 kBq Am-241 alpha source, extracted from a smoke detector, with a diameter of 2.7 mm. This source was attached to a 3D-printed stand with a double side sticker. At the bottom of the stand was a metal gasket, which served as a controlled reflection surface.A 3 MBq Am-241 alpha source with a diameter of 12.5 mm. It was placed at the center of a 3D-printed stand. No reflection surface was attached as there was a metal ring around this source, which served as a controlled surface.

Both sources are depicted in Figure 2.

### 2.2. Large Lens Detection System

Two distinct alpha-radiation detection systems have been developed: a large lens detection system and a specialised system for imaging within glove boxes.

The large lens detection system is designed to capture alpha-induced radio-luminescence (RL) over long distances, leveraging its high resolution and sensitivity. This system’s capabilities are crucial for broader environmental assessments where precision and range are paramount.

#### 2.2.1. Camera

The detector used for the large lens detection system was a deep-cooled iKon-M 934 (Oxford Instruments Andor, Belfast, UK) camera. It is characterised by:Resolution: 1024×1024 pixels, each of size 13 μm × 13 μm, 8 × 8 pixel binning was used during the experiment to reduce the read noise per pixel; therefore, the modified resolution is 128×128 pixels.The quantum efficiency (QE), which is the measure of the effectiveness of an imaging device to convert incident photons into electrons, is 57% on average within the 300–400 nm range (refer to Figure 3).

#### 2.2.2. Optical Configuration

Two lenses were combined to configure a simple telescope:27-513 Condenser Lens (Edmund Optics, York, UK) made from N-BK7 glass: 200 mm diameter, 400 mm focal length, and approximately 70% transmission at 337 nm.LA4545-UV Fused Silica Plano-Convex Lens (Thorlabs Ltd., Lancaster, UK): 50.8 mm diameter, 100 mm focal length, AR Coating for the 245–400 nm range ensuring minimal reflection, and about 95% transmission at 337 nm.

The field of view is 10 degrees. The transmission properties of both N-BK7 and UV fused silica are based on data sourced from [30,31]. To facilitate precise focusing, the condenser lens was mounted on linear rods equipped with ball bearings. This setup allowed for smooth adjustments. Additionally, a stepmotor was integrated with a threaded rod to modify the position of this lens. By altering the lens’s position, the focus point of the system can be adjusted to ensure optimal image quality. Additionally, a 3D-printed filter wheel system was built, and affixed directly to the camera, facilitating efficient filter changes. The system’s motor mechanism, governed by an Arduino Uno (Arduino, Ivrea, Italy.) and A4988 Stepper Motor Drivers (AZDelivery Ltd., London, UK), permitted precise lens and filter positioning. The overview of the whole system is shown in Figure 4.

In addition to open-air environments, scenarios where alpha detection is required in confined spaces, such as gloveboxes used in nuclear facilities are considered. The glovebox imaging system, described subsequently, is tailored to address the unique challenges posed by these environments, including limited space and the need for external power supply.

### 2.3. Glovebox Imaging System

To accommodate the constraints of the glovebox’s portal dimensions, a more compact imaging system was adopted for testing within the glovebox.

#### 2.3.1. Camera

The Newport 78236 CCD (MKS Instruments, Newport, CA, USA) was employed, with its quantum efficiency illustrated in Figure 3. Originally having a resolution of 1024×512 pixels, each of size 26 μm × 26 μm. A 4 × 4 pixel binning was applied during the experiment, resulting in a resolution of 256×128.

#### 2.3.2. Optical Configuration

A simple telescope was configured using two lenses:A LB4592-UV 2-inch diameter UV fused silica lens (Thorlabs Ltd., Lancaster, UK) with a focal length of 60 mm.LA4148-UV (Thorlabs Ltd., Lancaster, UK), also crafted from UV fused silica, with a 1-inch diameter and a focal length of 50 mm.

The field of view is 13 degrees.

#### 2.3.3. Power Supply

The Jackery Explorer 240 portable power supply (Jackery, Oakland, CA, USA) was utilised to energize the camera within the glovebox. It has an operational duration of approximately four hours.

#### 2.3.4. Glovebox

The glove box, measuring 3 m × 0.75 m × 1 m, was procured from NIS Ltd., Reading, UK. The glove box and the entire equipment assembly are shown in Figure 5.

The glovebox features 12mm thick acrylic windows, which effectively block most of the UV light from sunlight outside the glovebox. To assess the UV-blocking efficacy of acrylic windows in relation to their thickness, a series of tests were conducted using a collection of round acrylic windows. Each window in the set was 1.7 mm thick and 50 mm in diameter. The test involved sequentially stacking these acrylic windows and measuring their cumulative ability to block UV light.

In practice, one acrylic window was positioned at a time in front of the detection system’s lens, gradually adding more windows to the stack. After the installation of each additional window, the system’s UV light detection was measured to ascertain the extent to which the increasing thickness of acrylic impacted the transmission of UV light.

### 2.4. Filter Selection

Throughout the experiments, a variety of optical filters was used to reduce ambient light background. The details of these filters are summarised in Table 1. Figure 6 displays the transmission properties of these filters.

### 2.5. Light Intensity Measurements in Different Environments

Considering the variability in ambient light intensity across different environments, it is crucial to understand the blocking rate of filters required to manage the ambient light background. This understanding is vital for selecting appropriate filter combinations. To evaluate the ambient light-blocking capacity, a stack of neutral density (ND) filters NDUVW40B series (Thorlabs Ltd., Lancaster, UK) was employed.

### 2.6. Image Processing and Analysis

Image exposures typically span several minutes. The raw images were converted from the proprietary format using Andor SOLIS software (version 4.30.30024.0) to CSV files. During these prolonged exposures, cosmic rays and gamma rays from the Am-241 present significant sources of noise, often manifesting as intense peaks in individual pixels. To mitigate the impact, a median filter with a 3×3 kernel (Python 3.11.9, scipy.ndimage package) was applied to the captured images.

Post-processing, the RL signal image was represented using a colormap and superimposed on a grayscale visible light image (captured using the FBH450-40 450 nm filter) to indicate the alpha source’s location. Image plotting was performed using Python 3.11.9 and the matplotlib package.

## 3. Results

### 3.1. Blocking Rate Required to Suppress Ambient Light

CCD cameras exhibit a broad spectral response, spanning from 200 nm to 1100 nm. Given the relatively low intensity of alpha RL, even minimal light sources, such as indicators on a power supply, can inundate the desired signal in the absence of a filter. Nonetheless, even with a filter in place, complete suppression of ambient light might not be achieved. Stacked ND filters were employed to gauge the level of ambient light in various settings. The minimum optical density (OD) required to bring the ambient light background below 1 electron/pixel/minute was determined. In such scenarios, the principal noise source would be the camera itself. These findings are presented in Table 2.

### 3.2. Imaging of Alpha RL under LED Room Light

In the experiments conducted under room light conditions, a significant challenge was to filter out intense ambient light effectively to discern the weak signal of alpha-induced RL. As identified earlier, the required blocking rate to mitigate room light is approximately OD11. However, the hard-coated 337 nm interference filters from Edmund Optics had an average blocking rate of OD4. To bridge this gap, a method of stacking filters was adopted.

Interference, or dichroic filters, operate by reflecting undesired wavelengths while transmitting the target wavelengths. The primary complication in stacking these filters is that the cumulative blocking rate does not linearly increase due to light reflections between the stacked filters, which can diminish their overall blocking effectiveness.

To overcome this limitation, a ‘sandwich’ filter structure was utilised as shown in Figure 7, incorporating an absorptive filter Hoya U340 (UQG Ltd., Cambridge, UK) between two interference filters (337 nm). This absorptive filter absorbs reflections between the interference filters, thereby enhancing the total blocking rate without significantly impacting the transmission of desired wavelengths.

Figure 7 visually represents this innovative filter arrangement. The diagram contrasts the multi-reflection issue in standard stacked dichroic filters with the ‘sandwich’ configuration, emphasising how the latter effectively mitigates internal reflections to enhance the blocking rate.

Employing this ‘sandwich’ structure in the experiments led to a blocking rate exceeding OD11, while still maintaining a 67% transmission rate within the signal wavelength band. Consequently, this approach proved highly effective in filtering out room light, allowing accurate imaging of alpha RL. Notably, during a 1-h exposure under bright LED room light, the interference from room light was reduced to less than 1 count per pixel per hour, a testament to the efficacy of the filter configuration.

Two distinct sources were examined in the experiments under bright LED room light using the large lens system. The first source, a 29 kBq sample, was studied under two different exposure times. A 10-min exposure revealed a clear alpha RL signal, as depicted in Figure 8. Extending the exposure time to 5 h brought to light a corona spanning 15 mm in diameter. The diameter of the corona exceeds that of the alpha source itself. This phenomenon can be attributed to the fact that the alpha particles traverse a short distance in the air, thereby inducing alpha RL in the vicinity of the source.

The 3 MBq alpha source was also tested. At 3 m distance, this source and the corona around it can be clearly seen in 1 min, as shown in Figure 9. It is worth noticing that because this source is a 12 mm diameter source, in the middle of the RL image the intensity of alpha RL is high (represented by the orange colour) and relatively even distributed within the diameter of the source. Outside the source, the intensity of the alpha RL gradually decreases.

The alpha RL signal is proportional to the activity of the alpha source and inversely proportional to the square of the detection distance [32]. Based on that, a 3 MBq alpha source can be detected at a 30 m distance with an exposure time of 10 min.

Additionally, the performance of the 337 nm imaging system was assessed under other light sources. While the system operated effectively under LED lighting, it faced challenges under fluorescent lights and incandescent lamps. This limitation is due to the fact that these light sources emit radiation significantly below the 337 nm threshold [33]. However, the system’s functionality remained unaffected when a fluorescent light was equipped with an acrylic cover, which effectively blocked most of the UV emissions from the fluorescent light.

### 3.3. Separation of Two Alpha Sources

The ability of the large lens system to differentiate between a faint alpha source positioned in close proximity to a strong alpha source was tested. The 3 MBq and 29 kBq alpha sources, as shown in Figure 10, were placed near each other at varying distances. The sources were imaged together at a 2 m distance with a 1 h exposure time. The experimental setup utilised the sandwich filter combination operating at 337 nm, conducted under LED lighting conditions.

Figure 11 displays the outcomes of the experiment. At a separation distance of 6 cm, the RL signal from the 29 kBq source is overshadowed by the strong RL signal from the 3 MBq source. As the separation increases to 7 cm, the RL signal from the 29 kBq source becomes perceptible. Beyond an 8 cm separation distance, the RL signal from the 29 kBq source is distinctly discerned from the RL signal emitted by the 3 MBq source. Hence, the minimum separation distance required for detecting a 29 kBq source against the background of a 3 MBq source at a 2 m detection distance is approximately 7 cm.

### 3.4. Imaging of Alpha RL Inside a Glovebox

Previous sections discussed the UV-blocking properties of acrylic. To empirically evaluate this characteristic, multiple 1.7 mm thick acrylic windows were stacked and positioned in front of the lens of the glovebox imaging system. The camera was equipped with the 337 nm filter system, which transmits light between 332 nm and 342 nm. The testing occurred under conditions of direct sunlight on a cloudy day, with the resulting data presented in Table 3.

This set of data can be fitted by the following equation:(2)log(C)=−0.4476×d+6.8136
where C is the Read Count per Pixel per Minute, d is the thickness of acrylic in mm. From this equation, the counts can be represented as a function of acrylic thickness:(3)$$C=6.5\times 10^{6}\times 0.357^{d}$$

Therefore, the transmission rate of 1 mm acrylic in 332–342 nm is approximately 35.7%. For a window 10 mm thick, the transmission rate is $3.4\times 10^{-5}$, corresponding to a blocking rate of about OD5. This level of attenuation underscores the effectiveness of acrylic in blocking UV light, particularly relevant for imaging applications within a glovebox.

This quantitative analysis confirms the suitability of acrylic as a UV filter in environments with high UV exposure. By utilising acrylic windows of appropriate thickness, the UV interference can be effectively minimised, thereby enhancing the accuracy and reliability of the detection system.

Building on these findings, the glovebox imaging system equipped with the 337 nm filter system was installed inside a glovebox as shown in Figure 5. The glovebox was placed in an environment with sufficient sunlight. The UV background within the glovebox was found to be less than 1 count per pixel per minute, a level low enough to be considered negligible.

Subsequently, two alpha sources were examined inside the glovebox. Figure 12 showcases the results, illustrating the alpha RL signals overlaid on visible light images for a clear representation of the sources. The lower images display the overlaid alpha RL signals, while the upper images serve as visual references captured with a conventional camera. On the left, a 3 MBq alpha source with a 5-min exposure at 3 m is displayed, and on the right, a 29 kBq alpha source with a 15-min exposure at 15 cm is presented.

The 3 MBq alpha source was placed at different distances from the camera. The peak signal, which is the average signal count per pixel within the area of the source, was recorded. The variations in the peak signal with respect to the distance between the alpha source and the camera were plotted in Figure 13.

## 4. Conclusions

This comprehensive study has successfully demonstrated new regimes of sensitivity for detecting alpha RL from radioactive sources. The system’s robustness under various lighting conditions, particularly with LED illumination, is notable. The challenge encountered under fluorescent and incandescent lighting due to their deep UV emissions highlights the importance of selecting appropriate lighting conditions or modifying them to fit the system’s operational range.

The innovative use of stacked filters, incorporating a sandwich structure with absorptive filters placed between interference filters, significantly improved the blocking rate, achieving over OD11. This configuration allowed for effective suppression of ambient light while maintaining high transmission in the signal wavelength band. By using this system, a 29 kBq source was detected at 3 m in 10 min. The system holds the capability to detect a 3 MBq source at a distance of 30 m in a similar 10-minute interval. The spatial accuracy of the system was further validated by its ability to detect a 29 kBq source positioned just 7 cm away from a 3 MBq source at a 2-m distance.

Furthermore, this study has successfully demonstrated the long-range alpha detection ability inside the glovebox, where acrylic’s UV-blocking properties significantly mitigated sunlight interference, confirming the system’s operational viability under varied lighting conditions.

In summary, the findings represent a substantial advancement in remote alpha monitoring, with direct applications in nuclear safety, forensics, and environmental protection, paving the way for safer and more efficient radiation detection methodologies.

## Figures and Tables

**Figure 1 sensors-24-03781-f001:**
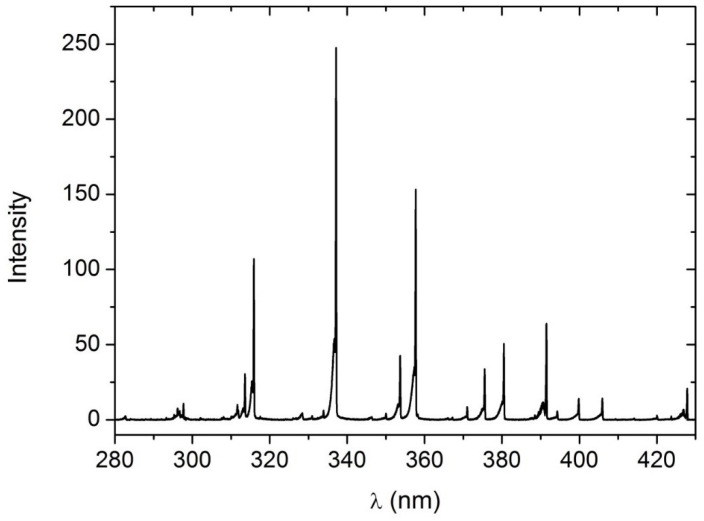
Spectrum of alpha-induced RL from artificial air (80% N_2_, 20% O_2_) at 800 hPa [14].

**Figure 2 sensors-24-03781-f002:**
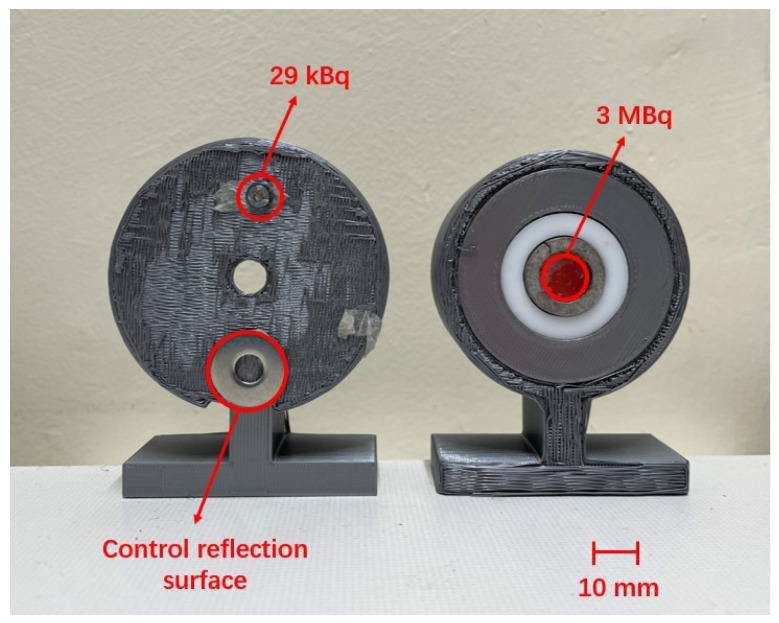
Alpha sources: 29 kBq and controlled reflection surface (**left**), 3 MBq (**right**).

**Figure 3 sensors-24-03781-f003:**
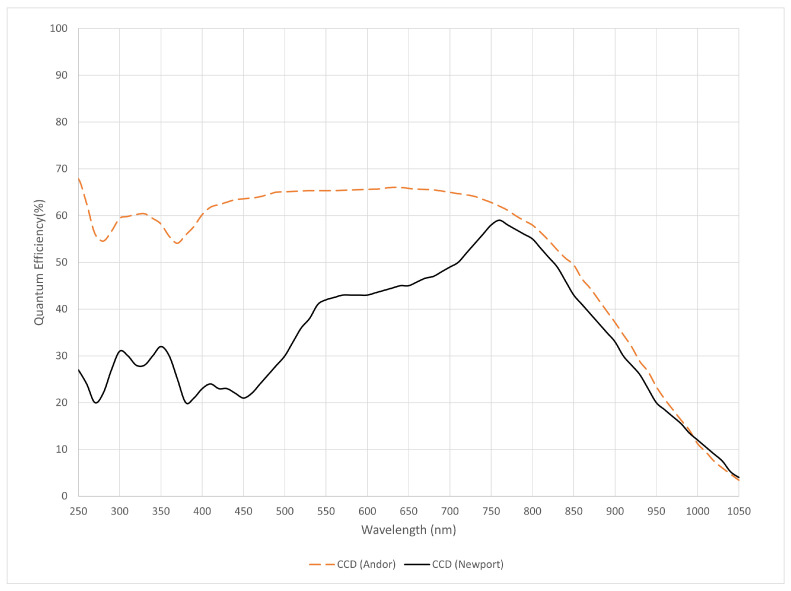
Quantum efficiency of the Andor iKon-M CCD and the Newport CCD.

**Figure 4 sensors-24-03781-f004:**
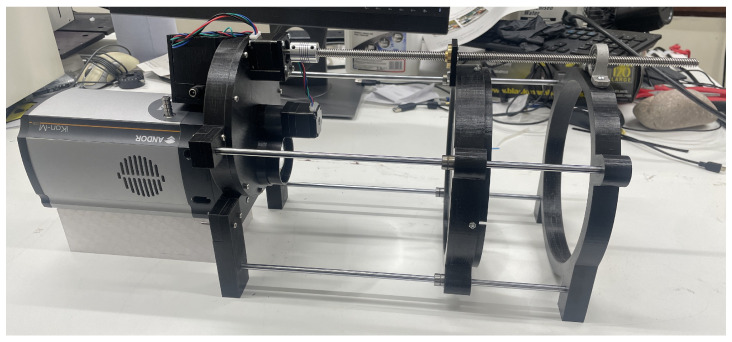
Overview of the large lens detection system.

**Figure 5 sensors-24-03781-f005:**
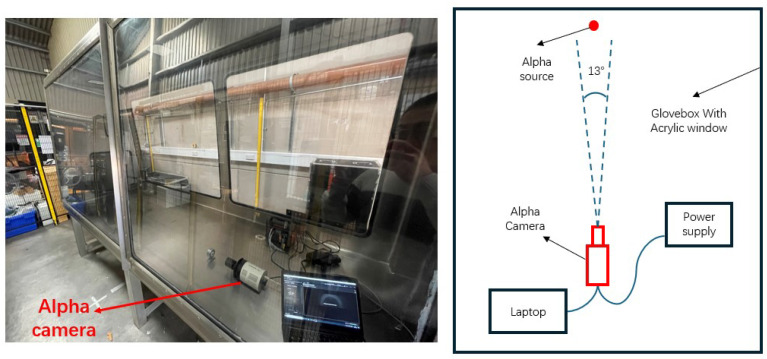
Overview of the glovebox and experiment setup inside the glovebox.

**Figure 6 sensors-24-03781-f006:**
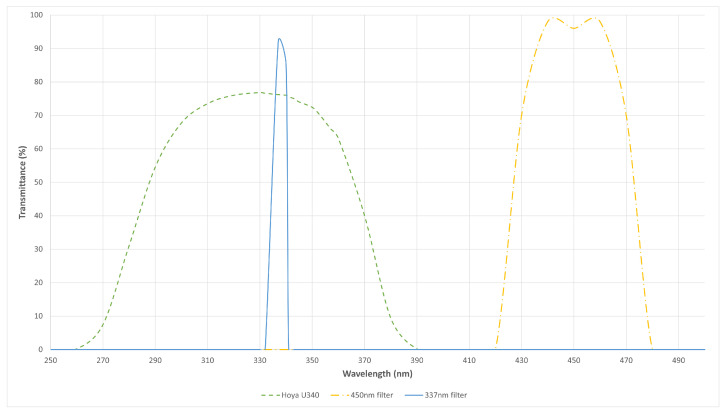
Transmission profiles of the various filters.

**Figure 7 sensors-24-03781-f007:**
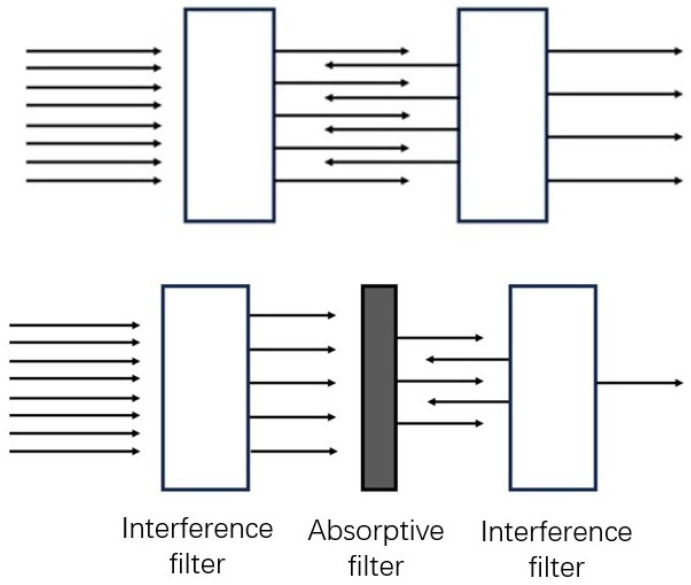
The upper image illustrates multi-reflections occurring between two stacked interference filters, reducing their overall blocking capability. The lower image depicts the ‘sandwich’ filter structure, showing the strategic arrangement of interference and absorptive filters to optimize ambient light blocking.

**Figure 8 sensors-24-03781-f008:**
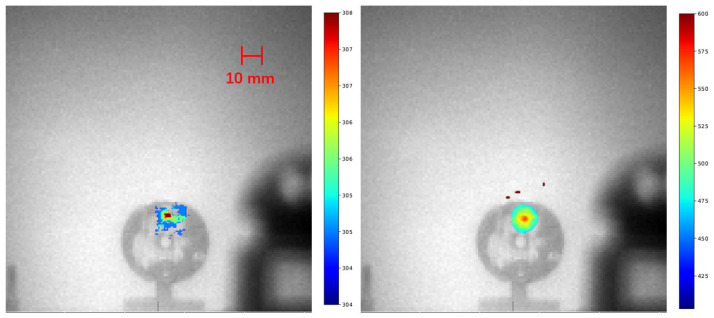
Alpha RL signal emanating from a 29 kBq alpha source positioned 3 m away. The images show results from exposure times of 10 min (**left**) and 5 h (**right**).

**Figure 9 sensors-24-03781-f009:**
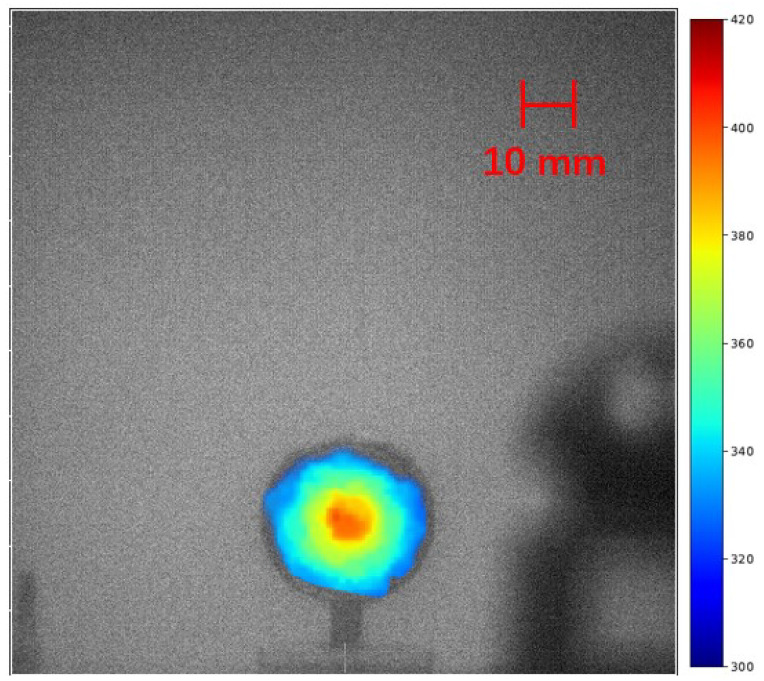
Alpha RL signal emanating from a 3 MBq alpha source at 3 m distance, with an exposure time of 1 min.

**Figure 10 sensors-24-03781-f010:**
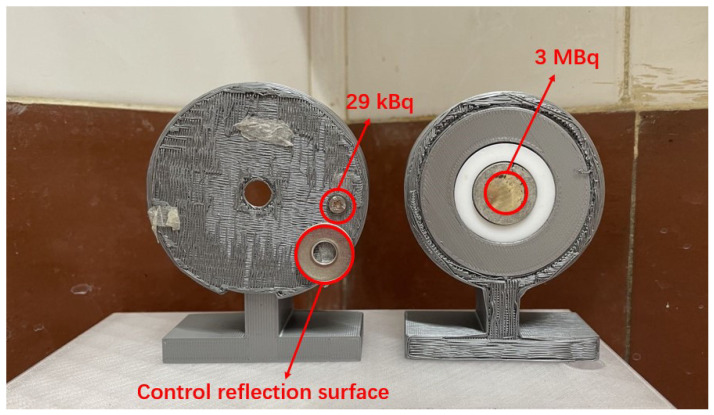
29 kBq source and the gasket as the controlled surface (**left**), 3 MBq source (**right**).

**Figure 11 sensors-24-03781-f011:**
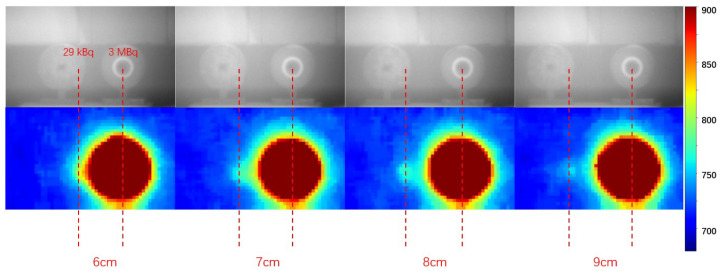
Visible light images from the 450 nm filter of the 29 kBq and 3 MBq alpha sources placed in close proximity at various distances (**top**). The corresponding RL signals captured using the 337 nm filter at a 2 m distance with a 1 h exposure time (**bottom**).

**Figure 12 sensors-24-03781-f012:**
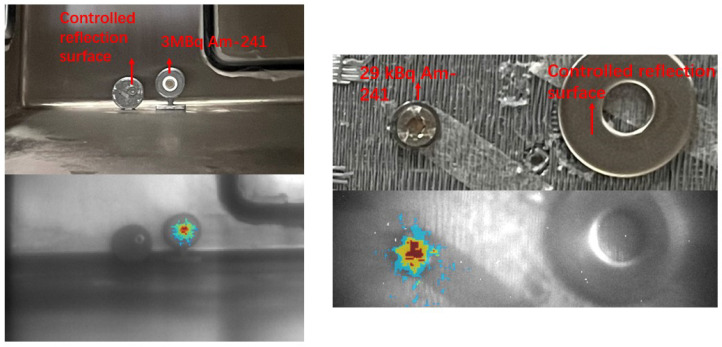
Overlay of alpha RL signals on visible images (**bottom**) compared with reference images taken by a conventional camera (**top**). Left: 3 MBq alpha source with a 5-min exposure at 3 m. Right: 29 kBq alpha source with a 15-min exposure at 15 cm.

**Figure 13 sensors-24-03781-f013:**
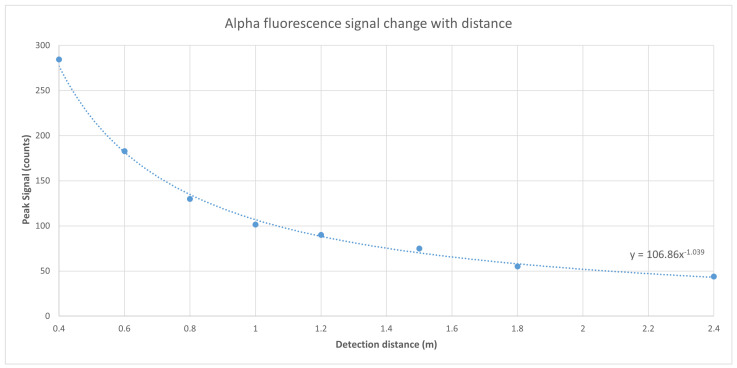
The peak RL signal from a 3 MBq alpha source change with detection distance.

**Table 1 sensors-24-03781-t001:** Filters utilised in the experiment.

Filter Name	Seller	Specification	Description
65–128	Edmund Optics, York, UK	CWL: 337 nm, FWHM: 10 nm, OD: 4	Detects alpha RL around 337 nm
Hoya U340	UQG Ltd., Cambridge, UK	Absorptive filter, bandpass: 275–375 nm	Additional filter to block ambient light
NEK01	Thorlabs Ltd., Lancaster, UK	Neutral-density (ND) filter kit	Used for ambient light measurement in various environments
FBH450-40	Thorlabs Ltd., Lancaster, UK	CWL: 450 nm, FWHM: 40 nm, OD: 5	Capture image in visible band for overlapping with alpha RL signal

**Table 2 sensors-24-03781-t002:** Blocking rates needed to suppress ambient light across different environments.

Light Environment	Primary Light Sources	Blocking Rate Required
Dark Room (Inside)	Power supply indicators	<OD4
Low Light Room (Inside)	Computer screen, emergency escape light	OD7
Night Time (Outside)	Moonlight, city lights	OD9
Room Light (Inside)	LED or fluorescent light	OD11
Daytime Shade (Outside)	Sunlight	OD13
Indirect Sunlight (Outside)	Sunlight	OD14
Direct Sunlight (Outside)	Sunlight	>OD15

**Table 3 sensors-24-03781-t003:** Background signal from sunlight for the 337 nm filter system, measured with varying acrylic window thickness in front of the lens.

Number of Acrylic Windows Stacked	Acrylic Thickness	Read Count per Pixel per Minute
1	1.7 mm	1,320,000
2	3.4 mm	234,000
3	5.1 mm	30,000
4	6.8 mm	4080
5	8.5 mm	900
6	10.2 mm	180
7	11.9 mm	40

## Data Availability

Data are contained within the article.

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
