# Peer review of "Advancements in Remote Alpha Radiation Detection: Alpha-Induced Radio-Luminescence Imaging with Enhanced Ambient Light Suppression"

_sensors, 2024, doi:10.3390/s24123781_

Round 1

Reviewer 1 Report

Comments and Suggestions for Authors

Very interesting article about non-conventional application of lumiescence phenomena to detection of radionuclides/radioactive sources. Well -designed and clearly described. The composition of manuscript is optimal.

Nevertheless, some minor issues occured:

line 9 incorrect  symbol: sholud be Am-241 or 241Am

Figure 5 - no reference in text. "Quantum efficiency" - please describe more detaily. Mayby "counting efficiency" is better term?

My recommendation is to publish after minor revision.

Author Response

Thank you for your insightful comments and the positive feedback on our article. We appreciate your recognition of the manuscript's design and clarity.

In response to the issues you highlighted:

  1. We have corrected the symbol error on line 9 to Am-241.

  2. Regarding Figure 5, we have provided a explanation of "quantum efficiency" in line 142: "The quantum efficiency (QE) of a CCD camera is a measure of its effectiveness in converting incident photons into electrons." This term is standard for CCD, but we considered your suggestion and clarified the context to avoid any ambiguity.

Please find the revised manuscript attached for your review. We hope that these modifications meet your expectations for minor revisions.

Reviewer 2 Report

Comments and Suggestions for Authors

It’s an interesting work on Remote Alpha Radiation Detection, my comments are as following:

1.     Figure 4. As an academic paper, Overview of the glovebox is not proper to show here, it’s a common goods in lab; and , the picture of experiment setup inside the glovebox is disorder and shows unclear information. A schematic diagram might be more proper.

2.     Figure 10, since the data is not original, it should be used as support information

3.     There should be a special section to show the experimental equipment and methods

4.     Although there are 14 figures and related discussions, I feel hard to follow the logic of the paper, I suggest to rewrite the paper according to the academic style

Author Response

Thank you for your constructive comments and the opportunity to enhance our manuscript on Remote Alpha Radiation Detection. We value your insights and have addressed each point as follows:

  1. Figure 4: We have replaced the photograph of the glovebox setup with a schematic diagram to clearly illustrate the experimental arrangement. 

  2. We agree with the comment and removed this picture, as the blocking rate of acrylic is discussed in section 3.4. Imaging of Alpha RL Inside a Glovebox

  3. Experimental Equipment and Methods: Our manuscript includes a detailed "Materials and Methods" section that we believe comprehensively covers the necessary information about the experimental equipment and methods used. We are open to expanding this section if further details are deemed necessary.

  4. Academic Style and Logic: We have revised several passages to adopt a more passive academic style, as suggested. However, we found this feedback somewhat broad and would greatly appreciate more specific guidance to ensure that our revisions meet your expectations. We aim to improve the logical flow and readability of the paper and would welcome any further advice you can provide.

Thank you again for your thorough review and helpful suggestions. We look forward to your further guidance to finalize the manuscript for publication.

Reviewer 3 Report

Comments and Suggestions for Authors

This manuscript contains valuable experimental results and information on methods and materials. Impacts of this paper might be limited, as remote alpha detection was demonstrated only in controlled environment and may not be applicable to practical cases. It is still appropriate for publication.

Author Response

Thank you very much for the positive comment. We are working on long range alpha imaging in open environment and hopefully will publish the results soon.